# A Multi-Working States Sensor Anomaly Detection Method Using Deep Learning Algorithms

**DOI:** 10.3390/s25185686

**Published:** 2025-09-12

**Authors:** Di Wu, Kari Koskinen, Eric Coatanea

**Affiliations:** Faculty of Engineering and Natural Sciences, Tampere University, 33720 Tampere, Finland; kari.koskinen@tuni.fi (K.K.); eric.coatanea@tuni.fi (E.C.)

**Keywords:** sensor anomaly detection, deep learning, LSTM, data-driven

## Abstract

The data collected from sensors are subject to the presence of anomaly data. These anomalies may stem from sensor malfunctions or poor communication. Prior to the processing of the data, it is imperative to detect and isolate the anomaly data from the substantial volume of normal data. The utilization of data-driven approaches for sensor anomaly detection and isolation frequently confronts the predicament of inadequately labeled data. In one aspect, the data obtained from sensors usually contain no or few examples of faults and those faults are difficult to identify manually from a large amount of raw data. Additionally, the operational states of a machine may undergo alterations during its functioning, potentially resulting in different sensor measurement behaviors. However, the operational states of a machine are not clearly labeled either. In order to address the challenges posed by the absence or lack of labeled data in both domains, a sensor anomaly detection and isolation method using LSTM (long short-term memory) networks is proposed in this paper. In order to predict sensor measurements at a subsequent timestep, behaviors in the preceding timesteps are utilized to consider the influence of the varying operational states. The inputs of the LSTM networks are selected based on prediction errors trained by a small dataset to increase the prediction accuracy and reduce the influence of redundant sensors. The residual between the predicted data and the measurement data is used to determine whether an anomaly has been identified. The proposed method is evaluated using a real dataset obtained from a truck operating in a mine. The results showed that the proposed network with the input-selection method demonstrated the ability to accurately detect drift and stall anomalies accurately in the experiments.

## 1. Introduction

In modern industries, more and more sensors are involved in different systems to monitor the performance of each component during operation for the purposes of control or safety. Sensors, however, may generate anomalous signals as well due to harsh environmental conditions, poor installation, inadequate maintenance and so on [1], which will influence the stability, reliability, and accuracy of the systems. In fact, sensor anomaly may even cause system failure or other safety issues. Therefore, sensor fault detection is also necessary to monitor the state of sensors.

Sensor anomaly detection methods can be classified into two categories: one is model-based methods, and the other is data-driven methods. Model-based methods, having been applied in power systems [2], health monitoring [3], and so on, construct computational models for the system to estimate the sensor values. By comparing the differences between the estimated values and the real sensor values, sensor faults can be detected. However, these methods require accurate simulation or analytic models for the system, which are usually very complex and time-consuming. Moreover, the models are usually application-dependent, which means when the monitored system changes, or even the working conditions change, a new model should be constructed. On the other hand, data-driven methods, which do not need exact knowledge of the monitored system, rely on historical data to estimate the performance of the sensors. Recently, different data-driven sensor anomaly detection methods have been developed, such as random forest (RF) [4], support vector machines (SVM) [5], and neural networks (NN) [6]. One of the main research directions is to use classification methods to detect faults by using different machine learning approaches. For example, Gao et al. [7] employed SVM to classify the residuals of each sensor to detect fault sensors in a binary mode. In those classification studies, labeled data, especially data including faults, are needed to train the classifiers to achieve an accurate prediction. However, it is usually not possible in the real world to manually label fault data from large amounts of raw data, or the fault data are difficult to obtain, which makes these classification methods difficult to apply.

To overcome the problem of lack of labeled data, multiple data-driven approaches are applied in sensor anomaly detection, relying only on correct sensor data. One aspect of the approaches is the autoencoder (AE). As an unsupervised learning method, AE can learn and extract hidden representations from raw data to provide an estimation of sensor measurements. Thus, AE is suitable for fault detection problems only with correct data. Under the concept of AE, different kinds of neural networks are employed to realize the estimation of the sensor measurements. In [8], an auto-associative neural network (AANN) is used as an autoencoder method to detect and isolate multiple sensor faults in a non-linear system. Instead of estimating sensor measurements for a certain time spot, some studies use different networks to estimate the sensor measurements during a period. A long short-term memory (LSTM) network, as a type of recurrent neural network, is used to build the AE to estimate the sensor measurements during a period and isolate failures with insufficient data conditions in different input feature spaces [9]. Jana et al. employed convolutional neural networks (CNN) to capture the features of the sensor signal plot for a period, which is used in the AE framework [10].

Different from AE which can be defined as a re-construction process, another group of approaches uses NNs to predict sensor measurements according to historical data [11,12]. Darvishi et al. employed two multi-layer perceptron (MLP) networks to work as an estimator and a predictor, respectively [13]. The differences between the results from the estimator and the predictor are used to detect sensor faults and to classify the fault categories. Due to their ability to learn sequential information, RNNs, especially LSTM and its variants, are widely used as predictors to predict sensor measurements and their values are used to compare with the measurements from the sensors to detect faults. In [14], an RNN is used to predict sensor measurements according to the previous timestep’s value, which is compared with the predictions from a feedforward neural network (FFNN) to detect failures.

Besides lacking data containing anomalous signals, another aspect of lacking labeled data is the working conditions. In some cases, the machine may not work in a certain condition, and different workloads may cause different behaviors in the sensor measurements, such as the problem tackled in this paper, where the sensors mounted on a truck working in a mine are studied. During operation, a truck may work in different states, such as stalls, idling, accelerating, breaking and so on, and it may work in different environments, like up-hill or down-hill. Different working states and different environments may lead to different sensor measurement behaviors. Additionally, the driver’s behavior is influenced by the prevailing working conditions, thereby affecting the operation of the truck. However, the working conditions and decisions from the drivers are not labeled or recorded. Therefore, the detection method should be able to work in multiple environments, which is not well-studied. In this case, LSTM networks, which have the ability to learn features from sequential data, are employed to predict sensor measurements according to the historical behavior of the sensors. Additionally, a continuous fault threshold is set up to make sure to filter the sudden changes in the working conditions. To increase the fault detection sensitivity and reduce the rate of fake detection, the length of the time window is also studied in this paper.

Another feature that needs to be considered in multi-sensor anomaly detection is the input of each LSTM network to predict the behavior of each sensor. In previous studies, the predictors or estimators usually use either all the sensors’ data or the same sensor data as input. Considering all sensor data may cause a redundancy of information, leading to a fake detection in a certain sensor. On the other hand, only using their own historical data will result in the loss of information from other sensors to judge the accuracy of the prediction. Take the case study in the paper as an example, where sensors are utilized to assess the condition of a truck’s power system, thereby highlighting the existence of physical relationships between sensor measurements. For instance, the vehicle’s velocity is contingent upon the operational speed of the engine and the selected transmission gear. The employment of physics relationships in sensor measurement prediction enables the elimination of the influence of redundant data from less important sensors. To capture the knowledge of the physics relationships, an input selection method is proposed and added in the sensor fault detection process to find the most related sensors.

In summary, an LSTM-based sensor anomaly detection and isolation method is proposed to detect anomaly sensors under variate working conditions and with unknown inputs from outside operators (i.e., workers or drivers). The LSTM network is used to extract knowledge from historical data and predict subsequent timestep measurements based on the near timestep data. An input selection method has been developed to select the input sensors’ data for the measurement prediction. This method aims to inherit the physics relationships between sensors and reduce the influence of redundant information from less important sensor measurements.

The rest of the paper is organized as follows: Section 2 introduces the sensor anomaly detection problem solved in this study. Section 3 presents the details of the fault detection methods. The experiment data are presented and introduced in Section 4. Section 5 summarizes the comparison results of the proposed method to illustrate the performance of the LSTM-based sensor anomaly detection method with input selection features. Section 6 discusses the results and provides the influences of the window size selection on the detection accuracy. Section 7 concludes the paper.

## 2. Problem Formulation

In this study, six sensors mounted on a truck working in a mine are considered, as shown in Table 1. All the sensors are the exiting sensors on the truck to monitor the power system. They are all digital sensors. The data were provided through Body builder manufacturer gateway in the truck and routed to the cloud by IoE-GW (Internet of Everything-GateWay). The sensor signals conform with SAE J1939-71 [15] definitions.

The states of the truck include stall, idling, accelerating, breaking, uniform motion, and so on. However, the exact state at a given time is not clearly labeled. Additionally, the sensor data are not labeled with normal or abnormal, either. Therefore, the problem in this paper is defined as a sensor anomaly detection and isolation problem in a multi-task working condition. Additionally, the sensor measurements exhibit a strong correlation with the behavior of the drivers and the external environment. The absence of such data hinders the capacity for accurate measurement prediction by sensors.

In this study, physics relationships exist between sensor measurements. The position of the acceleration pedal exerts a direct influence on the engine speed and engine fuel rate. On the other hand, the vehicle speed is determined by the combination of the engine speed and the transmission selected gear. Consequently, the interrelationships among disparate sensors can also be leveraged to detect anomaly measurement when incongruent behaviors are identified in multiple interconnected sensors.

## 3. Methods

Since the states of the truck varied during operation and the states were not labeled, the traditional multi-layer neural network was not suitable as they usually work for one certain case. In this case, an LSTM network considering sensor data in the previous timesteps was employed to overcome the issue of the unlabeled working state by assuming the truck state does not have sudden changes during operation. The overall process of the fault sensor detection and isolation is shown in Figure 1. The raw data were normalized to make sure that the inputs for the LSTM network were on the same scale to increase accuracy. Each sensor had its own LSTM network. To estimate the influence of the other sensors on the studied sensor, the selection of input sensors was performed for each sensor in the input selection block. The anomaly period was identified in the anomaly detection block through a comparison of the prediction data with the measurement data. The identification of the anomaly period was achieved through the integration of measurements from multiple sensors, with consideration given to the interrelationships that exist among them. In conclusion, for the period in which an anomaly was detected, the anomaly sensor(s) were isolated according to the residuals of each sensor. The following sections present the details of each block.

### 3.1. Normalization Block

The training and testing data were normalized into 0–1 range using Equation (1),(1)$$x_{norm}=\frac{x-lb}{ub-lb}$$
where x is the real sensor measurement and $x_{norm}$ is the normalized value. The maximum and minimum values for each sensor in the training set were selected as the upper (*ub*) and lower boundaries (*lb*) in the normalization. Note that the boundaries were selected from the training set; some of the sensor values in the testing sets may be out of the selected boundaries and the normalized values may be smaller than 0 or larger than 1, especially for the fault sensor data.

### 3.2. Input Selection Block

The relationships among different sensors are considered in the anomaly detection process by selecting input sensors of the LSTM network. Two sensor selection methods are proposed and tested in this paper. First is the selection method based on prediction errors. For a given sensor, all the possible combinations of input sets are enumerated. In this case, there were six sensors. The number of combinations of input is 6 + 15 + 20 + 15 + 5 + 1 = 62. Then, LSTM networks using different combinations of input were trained by a small amount of data. Then, the prediction error, presented by using mean square error, was estimated using another test dataset. The input set with the smallest error was selected for the given sensor. The process was repeated for all the sensors to find the input set for every sensor.

The second method is employing the correlation matrix to ascertain the most important input sensors for a specific sensor. The correlation matrix for the training dataset is calculated and if the correlation coefficient for a sensor pair exceeds a threshold, the two sensors are determined to have a strong relationship. The threshold can be selected according to the distribution of the correlation coefficients in the case. As a result, when constructing the prediction network for one sensor in the pair, the other sensor in the pair is considered an input sensor.

### 3.3. LSTM Network Block

Considering the problem tackled in this study, the state of the truck during the operation was not clearly labeled. A time-series prediction model is applied by considering the historical data in the previous timesteps to ignore the influence of the different states of the truck. An LSTM network is a kind of recurrent neural network which can be applied to time-series prediction problems [16]. By using an input gate, an output gate, and a forget gate to control the flow of information into and out of the cell, an LSTM network has the ability to provide a short-term memory for a long time. A network with two LSTM layers, one fully connected layer and one output layer has been constructed in this study. The structure of the network is shown in Figure 2.

As illustrated in Figure 2, the data in time window *Tw* is fed into LSTM layer 1 to generate a set of time-series data for the extraction of time-series features. LSTM layer 2 is responsible for making a prediction based on the captured feature with the objective of outputting a vector for the predicted timestep. After a fully connected layer and an output layer, the value of the sensor at the next timestep is predicted. To avoid overfitting, two dropout layers are added after the LSTM layer 1 and the fully connected layer. The dropout rate is set to be 0.5 for both. A linear function is used as the activation function for all the layers. The mean square error (MSE) is used as the loss function, and the Adam optimizer is used to train the networks. All the training and prediction are realized by using TensorFlow-2.10.0 library in Python-3.10.

### 3.4. Anomaly Detection Block

Instead of only considering the prediction error of a single sensor to detect the anomaly, a joint prediction error is used to identify the anomaly period, considering the relationships between measurements from different sensors. The square prediction error (SPE) [17] is used as an index to detect the period when fault appears. The SPE is defined as follows:(2)$$SPE=\sum_{j=1}^{m}{\left(x_{j}-{\tilde{x}}_{j}\right)}^{2}$$
where $x_{j}$ is the measurement from the j-th sensor; ${\tilde{x}}_{j}$ is the prediction value from the LSTM network; m is the number of sensors, i.e., six in this study.

As in [18], the confident limit of the training set can be calculated as(3)$$\delta_{\alpha }^{2}=\theta_{1}{\left(1-\frac{\theta_{2}h_{0}\left(1-h_{0}\right)}{\theta_{1}^{2}}+\frac{z_{\alpha }\sqrt{2\theta_{2}h_{0}^{2}}}{\theta_{1}}\right)}^{\frac{1}{h_{0}}}$$
at $\left(1-\alpha \right)\times 100\%$ confidence level. In this study, the confidence level is set to be 95%. $z_{\alpha }$ is the normal variable corresponding to the confidence level, i.e., $z_{\alpha }=1.96$. $\theta_{i},i=1,\;2,\;3$ can be calculated as(4)$$\theta_{i}=\sum_{j=1}^{m}\lambda_{k}^{i}$$
where $\lambda_{k}$ are the eigenvalues of the sample covariance matrix ***S***, which is defined as(5)$$S\left(j,k\right)=\frac{1}{n}\sum_{i=1}^{n}e_{ij}e_{ik}$$
where n is the number of the training samples, $e_{ij}$ is the residual value of the j-th sensor in the i-th training sample, i.e., $e_{ij}=x_{i,j}-{\tilde{x}}_{i,j}$.

Therefore, if the SPE value calculated for a given timestep is larger than $\delta_{\alpha }^{2}$, an anomaly is detected. Since there are prediction errors that appear in the LSTM network, the SPE value for some timesteps will be larger than $\delta_{\alpha }^{2}$. To reduce the sensitivity of these random prediction errors, a continuous threshold is checked to determine if it is an anomaly measurement, which means the anomaly is detected when the anomaly lasts for at least *tf* timesteps. The value of *tf* can be determined from the training set by finding the longest continuous anomaly appearing period. In this study, *tf =* 7.

### 3.5. Anomaly Isolation Block

When a continuous anomaly is detected through the previous block, the next step is to isolate the fault sensors. The mean absolute error (MAE) of each sensor in the fault period, i.e., ${MAE}_{j}=\frac{\sum_{i=1}^{Tf}\left|x_{i,j}-{\tilde{x}}_{i,j}\right|}{Tf},\;j=1,\ldots 6$ is calculated, where Tf is the length of the fault period. Given the mean and standard deviation of the absolute error for the training set are ${\mu e}_{j}$ and ${\sigma e}_{j}$ for the j-th sensor, if ${MAE}_{j}$ is out of the range [${\mu e}_{j}-{\sigma e}_{j},\;{\mu e}_{j}+{\sigma e}_{j}$], the sensor is determined to have fault in this period.

## 4. Experiment Data Preparation

Sensor data from 10 individual days were well-selected and used in this research and are designated as Day1 to Day10. Note that the data recorded on a single day is continuous, while the data from two separate days is not continuous. Furthermore, the duration of the recorded data on each day exhibits variability. Consequently, the data are entered into the algorithm on separate days. On a daily basis, the sensor data undergoes synchronization with a 1 s timestep, with all data recorded continuously during the designated recording period. During the experiment period, the truck’s power system functioned without any issues, and the driver performed normally during the period. The data were meticulously screened for the experiment. No anomaly sensor measurements were identified in the data from Day1 to Day9, but stall errors were marked on Day10. To demonstrate the efficacy of the proposed method in other types of anomalies, five periods of random drift errors were introduced to the data on Day9. In each period, the anomaly data recorded by the sensors were also random. Figure 3 shows the example of drift error in the vehicle speed sensor on Day9 and the stall error on the accelerate paddle position sensor on Day10. Table 2 and Table 3 show the anomaly sensors on Day9 and Day10 marked using (X), separately. The final column lists the percentage of the anomaly measurements in the total data for the specific day.

In this case, the 10-day data are divided into two groups, the first four-day data (Day1 to Day4) are used to train and validate the LSTM-based prediction models, and the latter six-day data (Day5 to Day10) are used to assess the performance of the proposed methods. In summary, the training and validation set contain 50,743 s, while the testing set contains 111,706 s data.

## 5. Experiment Results

The performance of the proposed method is tested in this section. First, the benefit of using input selection blocks is tested in terms of prediction accuracy and the anomaly sensor detection accuracy. The metrics root mean square error (RMSE) is used to illustrate the accuracy of the prediction by using different input sets. The true positive rate (TPR) and false positive rate (FPR), as in Equations (6) and (7), are used to illustrate the detection accuracy and sensitivity of the proposed method.(6)$$TPR=\frac{TP}{TP+FN}$$(7)$$FPR=\frac{FP}{FP+TN}$$

TP stands for the number of correctly detected anomalies, FP stans for the number of erroneously detected anomalies, TN stans for the number of correctly identified normal samples, and FN stands for the number of erroneously identified normal samples.

### 5.1. Comparison of Different Input Selection Methods

Four input selection settings are tested in this section. They are as follows:Using all the six sensors’ data as inputs for a given sensor detection, marked as 6-input,Only using the own data as inputs for a given sensor detection, marked as 1-input,Using prediction error to select input set for a given sensor detection, marked as Acc-input, andUsing correlation matrix to select input set for a given sensor detection, marked as Cor-input.

As illustrated in Table 4, the determined input set for each sensor anomaly detection is shown via accuracy metrics and correlation matrix, respectively. It can be found that there is always at least one piece of sensor data that is removed from the set of inputs considering either the prediction accuracy or the correlations between variable pairs. There are small differences between the results of using the two methods for selecting sensors. The reason is that selecting based on prediction accuracy considers all the variables in the input set, while selecting by a correlation matrix considers the correlation between pairs.

Frist, the prediction accuracy of the LSTM networks using different input sets were tested on the test days without anomalies (Day5 to Day8). The RMSE values for each sensor are shown in Figure 4. It can be found that employing the input selection block can improve the prediction accuracy of the LSTM networks. Due to using accuracy as the selection criteria, using Acc-input setting outperforms the Cor-input setting in terms of prediction accuracy.

Table 5 lists the identified anomaly period on each of the testing days. Both the 6-input and 1-input settings suffer from erroneous anomaly detections on the day without anomalies, such as Day5, Day7 and Day8. The main reason is that the predictions are not very accurate and sometimes they act strangely. Figure 5 shows the plot of measurement data and prediction data with the 1-input setting of Sensor1 (i.e., acceleration paddle position) around the erroneously detected period on Day5, while Figure 6 shows the plot of measurement data and prediction data with the 6-input setting of Sensor1 around the erroneously detected period on Day7. In Figure 5, since the 1-input setting only considers its own historical data, the prediction cannot follow the sharp changes accurately, which causes erroneous detection during the rapid changes period. On the other hand, the 6-input setting may involve noises from less related sensor measurements. This causes the inaccuracy of the prediction and the erroneous anomaly detection in Figure 6.

For the data containing multiple drift errors, i.e., the data on Day9, using the 6-input and 1-input settings erroneously detects anomalies compared to the ones using selected inputs. Table 6 presents the TPR and FPR values for each sensor. In the context of drift error detection, the anomaly detection accuracy is similar for different input sets. However, it has been observed that using 6-input and 1-input settings is more sensitive to detecting erroneously, compared to using selected inputs. Figure 7 gives one example of the anomaly period of Sensor 6 (vehicle speed) on Day9. All four input sets were capable of detecting drift anomalies since substantial discrepancies were observed between the prediction and the measurement data. Given the existence of drift anomalies on Sensors 4 and 5 (engine coolant temperature and transmission selected gear), the 6-input setting generated the most significant discrepancies between the predicted and the measured data. Additionally, due to the existence of the time window to capture the historical data in LSTM, using the 6-input setting still erroneously marked the anomaly after the fault disappeared.

With regard to the data from Day10 which exhibited multiple instances of stall error periods in the recorded data, the 6-input setting identified 16 fault periods; the 1-input setting detected three fault periods, and both the selected input settings detected seven fault periods. The utilization of the 6-input setting can currently detect the real anomaly period but also erroneously detected normal sensor measurements due to the same lower accuracy reason in other test days. On the other hand, since the 1-input setting is constrained to the internal historical data of the system, it is unable to discern the anomalous behavior exhibited by the dataset. The employment of the input selection block has been demonstrated to yield the most accurate detection outcomes to other input settings. Table 7 gives the details of comparison in accuracy and sensitivity considering the individual sensors on the stall faults detection. Among all four settings, the input set selected by using accuracy criteria (i.e., Acc-input) demonstrates the optimal performance in the stall anomalies detection scenario in both accuracy and sensitivity. Utilizing the 6-input configuration has been demonstrated to facilitate precise anomaly detection during the true anomaly period; however, it concomitantly exhibits an elevated rate of erroneous detection. Using 1-input setting is inadequate for accurately detecting stall anomalies. Figure 8 shows the plot of predictions and measurements for Sensor 1 (acceleration pedal position) during the anomaly period on Day10. It is demonstrated that the prediction of using the 1-input setting follows the stall anomalies, which in turn lead to missed detection. Although using the Cor-input setting also generates a prediction close to the measurement, the gap is already out of the tolerance calculated in Equation (3) that the anomaly period is detected correctly.

### 5.2. Comparison with Autoencoder Approaches

In this section, the proposed predictive-based method is compared with two autoencoder-based approaches, namely LSTM-VAE [19] and OmniAnomaly [20]. These two methods utilize LSTM and Gated Recurrent Unit (GRU) as the encoder and decoder, respectively. Table 8 and Table 9 present the TPR and FRP values for the two methods in the context of drift faults detection and stall fault detection, respectively. Compared to the results of the proposed method listed in Table 6 and Table 7, the autoencoder-based methods demonstrate slightly superior performances in the detection of drift faults. However, their efficacies are significantly diminished in the context of continuous stall fault detection. Figure 9 shows the comparison between the measurement and the prediction value from the two autoencoder-based methods. An analysis of the data reveals that the drift fault behavior can be captured by employing both methods during the anomaly period, showing a similar prediction as the proposed method shown in Figure 7. However, the autoencoder-based methods are ineffective in the identification of stall faults. As shown in Figure 10, the prediction results from the autoencoder-based methods align with the anomaly data, while there is a gap between the measurement and prediction when employing the proposed method shown in Figure 8. Since the autoencoder methods have inherent a weakness in accounting for the physical relationships between sensor measurements, such as the cause-effect relationships between acceleration paddle position and engine fuel rate, the stall faults cannot be distinguished in the test case.

## 6. Discussion

### 6.1. Computational Complexity Analysis

LSTM networks and fully connected dense networks are used in the sensor anomaly detection method. According to [16], the computation complexity of the prediction network can be presented as(8)$$\begin{array}{l} O({(K}_{1}H_{1}+K_{1}C_{1}S_{1}+C_{1}S_{1}I_{1}+H_{1}I_{1})+\\ (K_{2}H_{2}+K_{2}C_{2}S_{2}+C_{2}S_{2}K_{2}+H_{2}K_{1})+\\ ({(K}_{2}+1)D_{1})+\\ {(D}_{1}+1)) \end{array}$$
where $K_{1}$ and $K_{2}$ are the number of output units in LSTM layer 1 and layer 2, $C_{1}$ and $C_{2}$ are the number of memory cell blocks in LSTM layer 1 and layer 2, $S_{1}$ and $S_{2}$ are the size of the memory cell blocks in LSTM layer 1 and layer 2, $H_{1}$ and $H_{2}$ are the number of hidden units in LSTM layer 1 and layer 2, $I_{1}$ is the number of units forward connected to memory cells, gate units and hidden units in LSTM layer 1, and $D_{1}$ is the number of output nodes in fully connected layer. Each line in Equation (8) shows the computational complexity of each layer in Figure 2. Since the number of weights for an LSTM network and a fully connected network can be calculated as(9)$$W_{LSTM}=KH+KCS+CSI+2CI+HI$$
and(10)$$W_{Dense}=(J+1)D$$
respectively, where J is the number of input nodes for the fully connected network, the computational complexity of the prediction network can be presented as(11)$$O(W_{LSTM1}+W_{LSTM2}+W_{Dense1}+W_{Dense2})$$
where $W_{LSTM1}$, $W_{LSTM2}$, $W_{Desne1}$, $W_{Desne2}$ are the number of weights in each of the layers in Figure 2, respectively. In this case, the number of weights for the prediction network is shown in Table 10. As the other blocks of the proposed sensor anomaly detection method are forward calculations, they have a fixed contribution to computational complexity. Therefore, the computational complexity of the proposed method is(12)$$O(W_{LSTM1}+W_{LSTM2}+W_{Dense1}+W_{Dense2}+1)$$

The networks are trained and tested on a i7-1370P CPU with 32 GB RAM. The average inference time for one sample is 6 ms.

### 6.2. Influence of the Window Size

The window size is one of the key hyperparameters in the proposed LSTM-based anomaly detection method. Although the LSTM demonstrates relative insensitivity to the window sizes in comparison with other RNNs, the window size exerts an influence on the prediction. A long window size may lead to an over detection at the end of continuous anomalies when too much anomaly sensor data are used in subsequent timestep predictions. On the other hand, a short window size may cause the detector to be too sensitive to a rapid change but insensitive for a long period anomaly.

In order to examine the impact of the window size, a series of tests with window sizes of 1 s, 5 s, 15 s, and 20 s was conducted, and the results were compared with using a window size of 10 s. For all four tests, the input selection block utilizes prediction accuracy as the selection criterion. Table 11 presents the anomaly periods detected by using different window sizes. No erroneous anomalies were detected on the normal dataset. In the context of the dataset containing drift faults, the efficacy of using different window sizes exhibits minimal variations. Table 12 gives the TPR and FPR values for the detection results by using different window size settings on the data with drift faults. As shown in the results, the window size exhibits minimal influence on the detection accuracy and sensitivity to the drift faults. On the other hand, the window size exerts a substantial influence on the detection of stall faults. As shown in Table 13, the detection network using 1 s, 15 s, and 20 s window sizes is incapable of detecting continuous stall faults from the provided dataset. Using a window size of 5 s performs similar to using 10 s as the window size like in Sensor 1, 2 and 3, but has a poor detection accuracy for Sensor 5. Therefore, the window size may influence the detection of the stall faults and window size = 10 has the best performance in the given case study.

## 7. Conclusions

To deal with the unlabeled data issue in data-driven sensor anomaly detection problems, an anomaly detection and isolation method based on an LSTM network is developed and applied to detect faults of the sensors mounted on a truck. The sensor data from previous steps are involved in adjusting the network to different working states of the truck to improve the prediction accuracy in the unlabeled working states environment. The residuals between the predicted sensor data and the measurement data are employed to distinguish fault signals considering the prediction errors. The comparison results demonstrate that the proposed method can detect anomalous data accurately with a high degree of accuracy, particularly in the context of stall fault detection compared to the autoencoder-based method. On the other hand, the window size exerts a substantial influence on detection of the continuous stall faults.

The future work will include adding a fault/anomaly classification to the detection and an isolation method to identify the resources of the fault/anomaly.

## Figures and Tables

**Figure 1 sensors-25-05686-f001:**
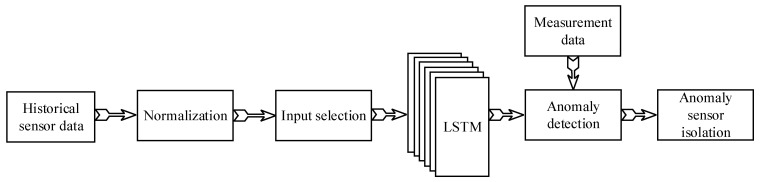
Process of fault detection and isolation.

**Figure 2 sensors-25-05686-f002:**
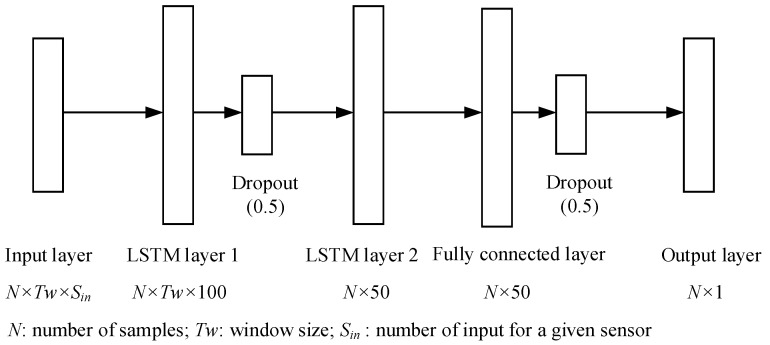
Structure of the proposed network.

**Figure 3 sensors-25-05686-f003:**
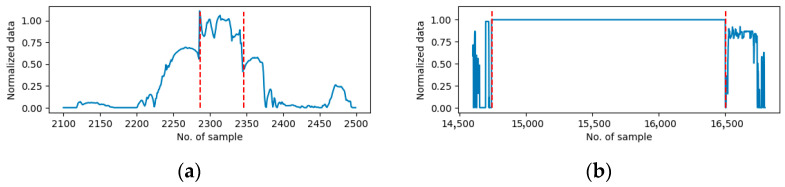
Examples of the anomaly sensor measurements shown in between two red dash lines. (**a**) Drift errors for vehicle speed sensors on Day9. (**b**) Stall errors for acceleration pedal position on Day10.

**Figure 4 sensors-25-05686-f004:**
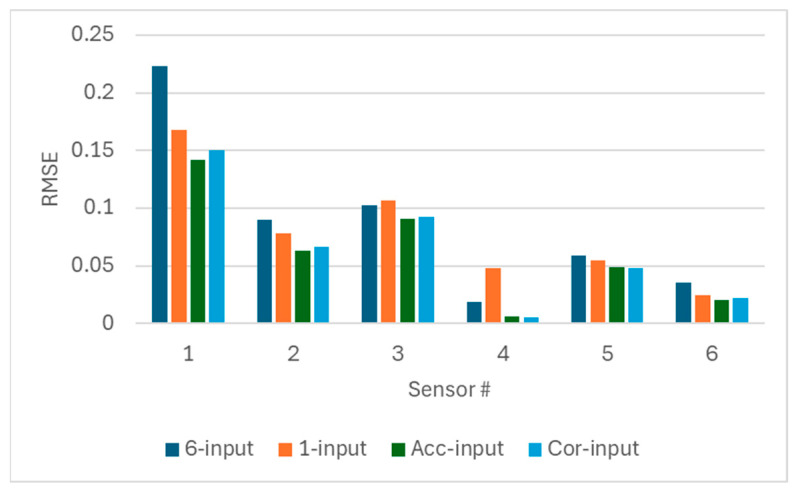
Comparison of prediction accuracy using different input sets.

**Figure 5 sensors-25-05686-f005:**
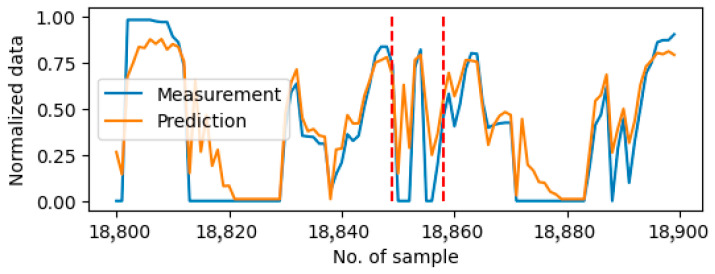
Erroneously detected anomaly using 1-input setting on Day5. The anomaly period is shown between two red dash lines.

**Figure 6 sensors-25-05686-f006:**
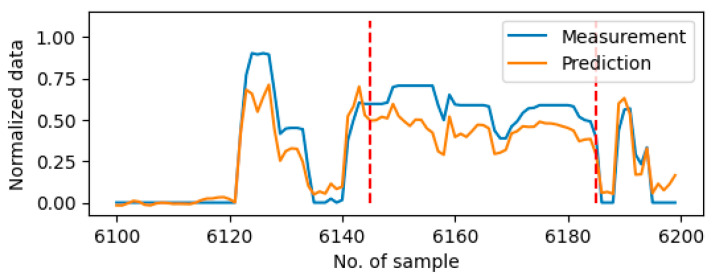
Erroneously detected anomaly using 6-input setting on Day7. The anomaly period is shown between two dash dotted lines.

**Figure 7 sensors-25-05686-f007:**
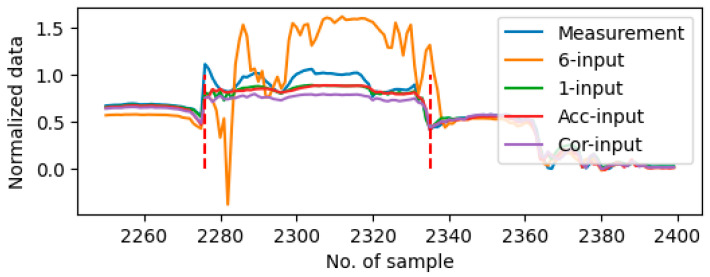
Plot of predictions using different input sets on Sensor 6 around the anomaly period on Day9. The anomaly period is shown between two red dash lines.

**Figure 8 sensors-25-05686-f008:**
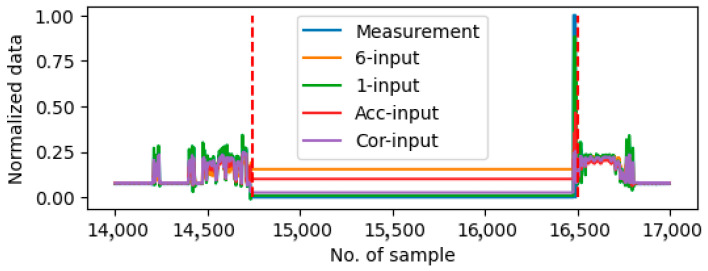
Plot of predictions using different input sets on Sensor 2 around the anomaly period on Day10. The anomaly period is shown between two red dash lines.

**Figure 9 sensors-25-05686-f009:**
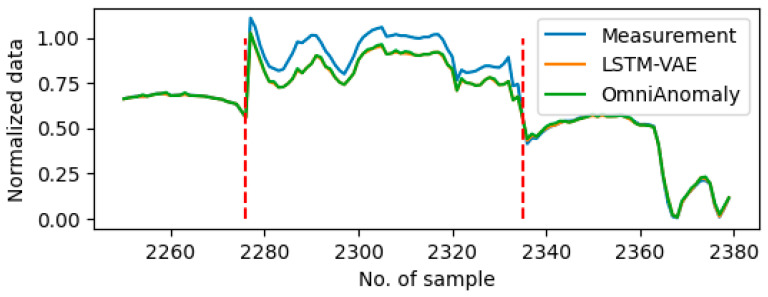
Plot of predictions using autoencoder-based method on Sensor 6 around the anomaly period on Day9. The anomaly period is shown between two red dash lines.

**Figure 10 sensors-25-05686-f010:**
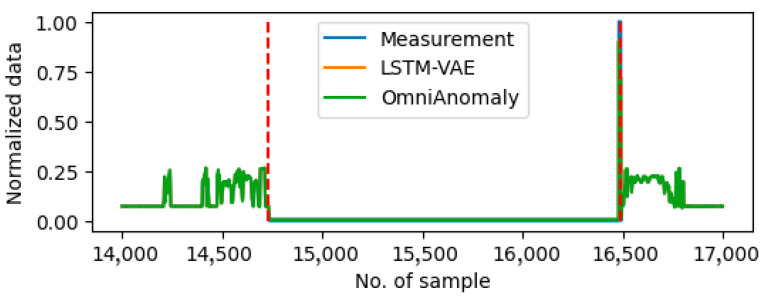
Plot of predictions using autoencoder-based methods on Sensor 2 around the anomaly period on Day10. The anomaly period is shown between two red dash lines.

**Table 1 sensors-25-05686-t001:** Sensors mounted on a truck.

Sensor #	Name
1	Acceleration pedal position
2	Engine speed
3	Engine fuel rate
4	Engine coolant temperature
5	Transmission selected gear
6	Vehicle speed

**Table 2 sensors-25-05686-t002:** Anomaly periods and percentage of anomaly data for each sensor on Day9. “X” shows the sensor which contains the anomaly data in the period.

	Period 1	Period 2	Period 3	Period 4	Period 5	Percentage
Sensor 1		X		X	X	1.9%
Sensor 2		X	X	X		1.5%
Sensor 3		X				0.3%
Sensor 4	X	X				0.9%
Sensor 5	X		X	X	X	2.8%
Sensor 6	X			X	X	2.2%

**Table 3 sensors-25-05686-t003:** Anomaly periods and percentage of anomaly data for each sensor on Day10. “X” shows the sensor which contains the anomaly data in the period.

	Period 1	Period 2	Period 3	Period 4	Period 5	Period 6	Period 7	Percentage
Sensor 1	X	X	X	X	X		X	13.4%
Sensor 2	X	X	X	X	X		X	13.4%
Sensor 3		X	X	X	X			9.3%
Sensor 4								0%
Sensor 5		X	X	X		X		8.5%
Sensor 6								0%

**Table 4 sensors-25-05686-t004:** Input sensors selected through the proposed method.

Sensor #	1	2	3	4	5	6
Acc-input	[1,2,3,5,6]	[1,2,5]	[1,3,5,6]	[1,3,4]	[2,3,5]	[1,6]
Cor-input	[1,2,4]	[1,2,5,6]	[2,3,5,6]	[1,4]	[2,3,5,6]	[2,3,5,6]

**Table 5 sensors-25-05686-t005:** Number of fault periods detected by different input selection settings.

	Day5	Day6	Day7	Day8	Day9	Day10
Anomaly period number	0	0	0	0	5	7
6-input	0	0	1	3	7	16
1-input	1	0	0	2	6	3
Acc-input	0	0	0	0	5	7
Cor-input	0	0	0	0	5	7

**Table 6 sensors-25-05686-t006:** Comparison of anomaly detection accuracy on Day9 with different input sets.

	6-Input		1-Input		Acc-Input		Cor-Input	
	TPR	FPR	TPR	FPR	TPR	FPR	TPR	FPR
Sensor 1	91.0%	1.4%	94.5%	1.6%	94.5%	0.3%	94.5%	0.3%
Sensor 2	96.9%	1.3%	42.8%	0.2%	97.1%	0.6%	93.9%	0.7%
Sensor 3	100%	2.1%	100%	0.8%	100%	0.9%	100%	1.6%
Sensor 4	100%	1.9%	98.9%	1.4%	100%	1.0%	98.9%	1.0%
Sensor 5	75.8%	0.6%	59.1%	0.2%	73.1%	0.2%	71.4%	0.1%
Sensor 6	71.8%	1.1%	26.4%	0.4%	76.8%	0.3%	76.8%	0.3%

**Table 7 sensors-25-05686-t007:** Comparison of anomaly detection accuracy on Day10 with different input sets.

	6-Input		1-Input		Acc-Input		Cor-Input	
	TPR	FPR	TPR	FPR	TPR	FPR	TPR	FPR
Sensor 1	91.7%	0.8%	0.4%	0.03%	93.8%	0.04%	90.0%	0.02%
Sensor 2	91.7%	0.7%	0.4%	0.02%	93.8%	0.02%	92.7%	0.03%
Sensor 3	95.0%	0.8%	0.5%	0.03%	98.2%	0.04%	92.5%	0.05%
Sensor 5	91.6%	1.00%	0.5%	0.03%	100%	0%	92.6%	0.05%

**Table 8 sensors-25-05686-t008:** Detection accuracy of LSTM-VAE and OmniAnomaly on the drift faults data.

	LSTM-VAE		OmniAnomaly	
	TPR	FPR	TPR	FPR
Sensor 1	95.0%	2.6%	96.1%	0.3%
Sensor 2	98.1%	2.8%	98.5%	0.6%
Sensor 3	97.0%	4.0%	100%	0.7%
Sensor 4	97.8%	3.6%	100%	1.0%
Sensor 5	88.4%	2.1%	89.4%	0.2%
Sensor 6	81.1%	2.4%	85.2%	0.3%

**Table 9 sensors-25-05686-t009:** Detection accuracy of LSTM-VAE and OmniAnomaly on the stall faults data.

	LSTM-VAE		OmniAnomaly	
	TPR	FPR	TPR	FPR
Sensor 1	51.7%	0.8%	80.1%	0.03%
Sensor 2	41.8%	0.7%	80.1%	0.02%
Sensor 3	33.1%	0.4%	85.5%	0.03%
Sensor 5	31.2%	0.7%	80.5%	0.03%

**Table 10 sensors-25-05686-t010:** Number of weights in the proposed method.

	Number of Weights
$W_{LSTM1}$	42,400
$W_{LSTM2}$	30,200
$W_{Dense1}$	2550
$W_{Dense1}$	51
Total	75,201

**Table 11 sensors-25-05686-t011:** Number of anomaly periods detected by using different window sizes.

Window Size (s)	Day5	Day6	Day7	Day8	Day9	Day10
1	0	0	0	0	5	6
5	0	0	0	0	5	6
15	0	0	0	0	7	12
20	0	0	0	0	7	9

**Table 12 sensors-25-05686-t012:** Comparison of anomaly detection accuracy on Day9 using different window size settings.

Window Size	1 s		5 s		15 s		20 s	
	TPR	FPR	TPR	FPR	TPR	FPR	TPR	FPR
Sensor 1	94.5%	0.8%	94.5%	0.5%	94.5%	0.6%	94.5%	0.3%
Sensor 2	86.9%	0.6%	95.0%	0.4%	85.1%	0.1%	70.2%	0.7%
Sensor 3	100%	1.0%	100%	0.9%	100%	1.0%	100%	1.6%
Sensor 4	96.8%	1.0%	98.9%	1.0%	98.9%	1.2%	76.4%	1.0%
Sensor 5	55.5%	0.3%	71.1%	0.2%	64.5%	0.2%	63.1%	0.2%
Sensor 6	55.9%	0.4%	73.6%	0.3%	56.8%	0.5%	66.8%	0.6%

**Table 13 sensors-25-05686-t013:** Comparison of anomaly detection accuracy on Day10 using different window size settings.

Window Size	1 s		5 s		15 s		20 s	
	TPR	FPR	TPR	FPR	TPR	FPR	TPR	FPR
Sensor 1	0.4%	0%	93.8%	0.01%	12.7%	0.1%	0.3%	0.05%
Sensor 2	0.6%	0.7%	93.8%	0.07%	12.7%	0.05%	0.6%	0.02%
Sensor 3	0.3%	0.6%	96.4%	0.07%	0%	0.08%	0.3%	0.01%
Sensor 5	0.5%	0.05%	2.2%	0.07%	0%	0.02%	0.6	0.07%

## Data Availability

The raw data supporting the conclusions of this article will be made available by the authors on request.

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
