# Peer review of "A Multi-Working States Sensor Anomaly Detection Method Using Deep Learning Algorithms"

_sensors, 2025, doi:10.3390/s25185686_

Round 1

Reviewer 1 Report

Comments and Suggestions for Authors

 a fault detection and isolation method using LSTM (long short-term memory)  
networks are proposed in this paper to detect faults of the sensors mounted on a truck’s power  system. Behaviors in the previous timesteps are used to predict the sensor measurements at the next timestep to consider the influence of the different working states. The inputs of the LSTM networks are selected based on prediction errors trained by a small dataset to increase the prediction accuracy  and reduce the influence of the redundant sensors. The residual between the predicted data and the measurement data is used to judge if it is a fault. The proposed method is tested by a real dataset obtained from a truck working in a mine and the impacts of input selection settings and window sizes are studied. As a result, the proposed network with the input-selection method and window size equaling to 10 can detect the sensor faults accurately. 

there are some  problems revised by authors:

1、there should be more comparisons to validate " the proposed input 427 
selection method based on prediction errors and window size = 10" in conclusion.

2、there are more explanations for figure 3 and figure 4. such as “selected inputs”;

3、the whole paper should be polished.

4、the  “. Comparisons and results ” should be more clearly.

Comments on the Quality of English Language

The whole paper should be described more clearly. Quality of English Language is good.

Author Response

Reviewer #1

a fault detection and isolation method using LSTM (long short-term memory)  
networks are proposed in this paper to detect faults of the sensors mounted on a truck’s power  system. Behaviors in the previous timesteps are used to predict the sensor measurements at the next timestep to consider the influence of the different working states. The inputs of the LSTM networks are selected based on prediction errors trained by a small dataset to increase the prediction accuracy and reduce the influence of the redundant sensors. The residual between the predicted data and the measurement data is used to judge if it is a fault. The proposed method is tested by a real dataset obtained from a truck working in a mine and the impacts of input selection settings and window sizes are studied. As a result, the proposed network with the input-selection method and window size equaling to 10 can detect the sensor faults accurately. 

there are some  problems revised by authors:

1、there should be more comparisons to validate " the proposed input
selection method based on prediction errors and window size = 10" in conclusion.

Response: Thanks for the comment. More comparisons have been added to the comparison section, including another input selection method based on correlation matrix and various window size as 1s, 5s, 15s, and 20s.

2、there are more explanations for figure 3 and figure 4. such as “selected inputs”;

Response: Thanks for the comment. To make the comparison and results section clearer, the section is rewritten by adding new comparisons and metrics to illustrate the performance. The figures are re-drawn to work as examples to show the performances. New explanations are added to the paragraph.

3、the whole paper should be polished.

Response: Thanks for the comment. The paper is revised carefully to increase the readability.

4、the  “. Comparisons and results ” should be more clearly.

Response: Thanks for the comment. As mentioned in the second comments. The section is re-written to make the comparison and conclusion clearer.

Reviewer 2 Report

Comments and Suggestions for Authors

Review the attached document

Author Response

Reviewer #2

Abstract: It is not entirely clear what type of sensor failure will be reviewed with the data obtained from the truck. It is also not mentioned how this will be differentiated from poor measurements or temporary disconnections, poor communication, etc., among the sensors. I believe it should be mentioned what type of sensor it is and whether it is analog or digital, or with a digital platform already integrated with a type of communication.

Response: Thanks for the comment. To make it clear, the phase “sensor fault” is replaced by “sensor anomaly”, which represents the measurement data are abnormal. This anomaly may be caused by sensor faults or by poor communication. To be specific, the data collected on Day 10 contains anomaly data due to poor communication. To test the performance of the proposed algorithm in detection and isolation other anomaly, random drifting faults are added to a data set without any anomaly. And this dataset is used to test the performance of the new method.

A brief introduction of the sensors is added in Section 2. All the sensors are the existing sensors originally on the truck. They are all digital sensors. The data was provided through Body builder manufacturer gateway in the truck and routed to the cloud by IoE-GW (Internet of Everything-GateWay). The sensor signals conform with SAE J1939-71 definitions.

  1. Problem Formulation

Table 1 should present the type of sensor used for each action of the truck and what type it is. It is not clear how it is guaranteed that the sensors will not fail in the first 9 days and that only on the 10th day will failures be recorded, since the type of sensor is not specified for each sensor.

Response: Thanks for the comment. The problem formulation section is re-organized. The sensors and data collection, transmission, and store processes are briefly introduced in this section to present the solved case. Additionally, the specific feature, i.e., the relationships between sensors’ measurement and the influence of the hidden inputs, such as environment and human operation, are also emphasized in the section to propose the hypothesis of using input selection method to improve the detection accuracy and reduce the influence from less important sensors in prediction.

An experiment data preparation section is added to summarize how to prepare data used in training, validating, and testing the method. Data collected in 10 days is used and the first four days are used to train and validate the network and the other six days are used in comparison. Note that the data are not necessarily collected from 10 continuous days. Separated data by day is because data in two days are not continuous in time. If connecting data in one dataset may cause training error when using LSTM. The data are checked and screened manually to make sure that only the data in Day 10 contains anomaly data caused by poor communication. To illustrate the performance of the algorithm, random shifting faults periods are added to the data in Day 9 to random sensors.

  1. Comparisons and Results

The graphs in Figure 3 do not have any units on the axes. It is also unclear why certain graphs with the sensors have certain predictive behavior. What difference does this make with the variation marked by the sensor response time?

Under this premise, it is assumed that the truck has no external conditions that affect it on the different days it travels its predicted route differently than the others.

Is the type of failure presented in Figure 5 for the sensors a period of total sensor disconnection? Regarding the computational cost of the data for the prediction, is a test performed to determine the critical failure time of a sensor in the truck's operation? Based on the above, how is it guaranteed that the failure is due to the sensor and not a communication problem or sensor variation in the environment?

Response:  Thanks for the comment. The x-axis of the plot is the number of samples, and the y-axis is the normalized sensor data. Thus, both axes are unitless. The different prediction behavior is caused by which sensor data are selected as inputs to train the model. Only using own data (one-input setting) causes low accuracy and using data from all sensors (6-input setting) may overfit when considering less relation sensors as the input.

There is no guarantee on the working routine, working conditions, and driver behaviors for all the experiment data. This is one of the main challenges in this case and other similar cases where the working conditions and the influence from the operators are not hidden. To select the experiment data, the guaranteed conditions are that there was no failure on the truck and the driver performed normally during the data recording period. On the other hand, there was no issues on the truck’s power system during the experiment period and the drivers behaved normally during the period.

The fault presented in figure 5 (now in figure 3(b)) is due to poor communication. Due to the lack of labels, the classification of resources of the failure is not discussed in this manuscript. It will be studied as future work.

Reviewer 3 Report

Comments and Suggestions for Authors

This paper presents a sensor fault detection based on LSTM. It suffers from crucial drawbacks including the lack of data and demonstration of applicability. The following concerns need to be addressed as well.

  1. How is it possible to claim that only 10-day data have generality? The data set in Table 1 is subject to the same concern. How can it be guaranteed that the proposed method works for other applications as well?
  2. Too many raw data are enumerated without sufficient analyses. The manuscript can be regarded as a case study, not a technical paper worthy of publication in the journal.
  3. The proposed scheme must be compared with the state-of-the-art works in the literature to highlight the advantages.
  4. The computational complexity should be analyzed and compared with the others to show the practicality.
  5. The specifications of the sensors in Table 1 must be provided.
  6. Why was the structure of the proposed network determined as shown in Fig. 2? The configuration must have be supported by proper reasoning.
  7. It would better to use term “measured” than “true” in the figures.
  8. It would be better to specify the dropout rate in Fig. 2.

Author Response

Reviewer #3

This paper presents a sensor fault detection based on LSTM. It suffers from crucial drawbacks including the lack of data and demonstration of applicability. The following concerns need to be addressed as well.

1. How is it possible to claim that only 10-day data have generality? The data set in Table 1 is subject to the same concern. How can it be guaranteed that the proposed method works for other applications as well?

Response:  Thanks for the comment. A data preparation section is added to explain how the data used in training, validation, and testing are prepared. The data contains normal measurement data and anomalous data due to poor communication. Additionally, random drift errors are added manually to the data on Day 9 in testing. Therefore, the authors believe that the experiment data can represent the performance of the method in anormal sensor detection and isolation in the case where the working conditions and input from operators are missing.

2. Too many raw data are enumerated without sufficient analyses. The manuscript can be regarded as a case study, not a technical paper worthy of publication in the journal.

Response: Thanks for the comment. A data preparation section (Section 4) is added to explain how the data used in training, validation, and testing are prepared. The selected case has its specific features that the working conditions of the truck and the behavior of the drivers are hidden. Therefore, the selected case can be a representation for the sensor anomaly detection under a certain condition.

3. The proposed scheme must be compared with the state-of-the-art works in the literature to highlight the advantages.

Response: Thanks for the comment. New comparison with autoencoder based methods has been added to the manuscript. As the results, the proposed method can achieve similar detection accuracy in the drift faults and performs much better in stall faults detection.

4. The computational complexity should be analyzed and compared with the others to show the practicality.

Response: Thanks for the comment. The analysis of the computational complexity is added in Section 6.

5. The specifications of the sensors in Table 1 must be provided.

Response: Thanks for the comment. All the sensors are the existing sensors originally on the truck. They are all digital sensors. The data was provided through Body builder manufacturer gateway in the truck and routed to the cloud by IoE-GW (Internet of Everything-GateWay). The sensor signals conform with SAE J1939-71 definitions.

6. Why was the structure of the proposed network determined as shown in Fig. 2? The configuration must have be supported by proper reasoning.

Response: Thanks for the comment. The first LSTM net is used to capture the time-series features for each sensor and the second LSTM net is used to summarize the features to one time step prediction with the assistance of the fully connect layers. Since the output of the prediction may contain less than -1 or larger than +1 values, linear activation functions are selected. A 0.5 dropout rate is set to avoid overshooting. The description is added to the section 3.3.

7. It would better to use term “measured” than “true” in the figures.

Response: Thanks for the comment. The term “true” is replaced by “measurement”.

8. It would be better to specify the dropout rate in Fig. 2.

Response: Thanks for the comment. The dropout rate is set to be 0.5 and added to figure 2.

Round 2

Reviewer 1 Report

Comments and Suggestions for Authors

All my concerns are ddressed.

Author Response

Thanks for your effort in reviewing the manuscript.

Reviewer 2 Report

Comments and Suggestions for Authors

Hello, good day. First of all, I appreciate the response to each of the comments made. I believe the answers were clear and elements were added that help to be more specific in the work done.

I also appreciate that the information added in each of the sections complements the work of the article. The choice of words to define "anomaly" and the inclusion of tables help understand the objective of the work.

The work is ready for publication. The wording and structure have been reviewed again and are correct.

Author Response

(The authors gave the same response as above.)

Reviewer 3 Report

Comments and Suggestions for Authors

The computational complexity of an algorithm proportional to the number of weights cannot be represented O(1), which is for fixed complexity. Besides, the number of weights is not enough to measure the complexity. More factors need to be considered.

Author Response

Thanks for the comment. The computational complexity of the proposed detection method has been revised in the manuscript and highlighted in red. In summary, the computational complexity is O(W_LSTM1+W_LSTM2+W_Dense1+W_Dense2+1), where, W_LSTM1, W_LSTM2, W_Dense1, and W_Dense2 are the number of weights in each of the layer in Figure 2, respectively. Since the other blocks in the proposed method are forward calculation, the contribution of other blocks to the computational complexity can be marked as O(1).